# In Vivo Verification of Treatment Source Dwell Times in Brachytherapy of Postoperative Endometrial Carcinoma: A Feasibility Study

**DOI:** 10.3390/jpm12060911

**Published:** 2022-05-31

**Authors:** Antonio Herreros, José Pérez-Calatayud, Facundo Ballester, Jose Barrera-Gómez, Rosa Abellana, Joana Melo, Luis Moutinho, Luca Tagliaferri, Ángeles Rovirosa

**Affiliations:** 1Fonaments Clínics Department, University of Barcelona, 08036 Barcelona, Spain; rabellana@ub.edu (R.A.); rovirosa@ub.edu (Á.R.); 2Radiation Oncology Department, Hospital Clínic Universitari, 08036 Barcelona, Spain; 3Radiation Oncology Department, Hospital Universitari i Politècnic La Fe, 46026 Valencia, Spain; perez_jos@gva.es; 4Radiation Oncology Department, Hospital Clinica Benidorm, 03501 Alicante, Spain; 5IRIMED, IIS-La Fe-Universitat de Valencia (UV), 46100 Burjassot, Spain; facundo.ballester@uv.es; 6Mathematics Department, Autonomous University of Barcelona, 08193 Bellatera, Spain; jbarrera@mat.uab.cat; 7NU-RISE LDA, PCI–Creative Science Park, 3830-352 Ilhavo, Portugal; joanasmelo@nu-rise.pt (J.M.); moutinho@nu-rise.pt (L.M.); 8U.O.C. Radioterapia Oncologica, Dipartimento di Diagnostica per Immagini, Radioterapia Oncologica ed Ematologia, Fondazione Policlinico Universitario A. Gemelli IRCCS, 00168 Rome, Italy; luca.tagliaferri@policlinicogemelli.it; 9Gynecologic Cancer Unit, Hospital Clínic Universitari, 08036 Barcelona, Spain

**Keywords:** brachytherapy, endometrial carcinoma, treatment verification, in vivo dosimetry, plastic scintillator dosimeter

## Abstract

(1) Background: In brachytherapy, there are still many manual procedures that can cause adverse events which can be detected with in vivo dosimetry systems. Plastic scintillator dosimeters (PSD) have interesting properties to achieve this objective such as real-time reading, linearity, repeatability, and small size to fit inside brachytherapy catheters. The purpose of this study was to evaluate the performance of a PSD in postoperative endometrial brachytherapy in terms of source dwell time accuracy. (2) Methods: Measurements were carried out in a PMMA phantom to characterise the PSD. Patient measurements in 121 dwell positions were analysed to obtain the differences between planned and measured dwell times. (3) Results: The repeatability test showed a relative standard deviation below 1% for the measured dwell times. The relative standard deviation of the PSD sensitivity with accumulated absorbed dose was lower than 1.2%. The equipment operated linearly in total counts with respect to absorbed dose and also in count rate versus absorbed dose rate. The mean (standard deviation) of the absolute differences between planned and measured dwell times in patient treatments was 0.0 (0.2) seconds. (4) Conclusions: The PSD system is useful as a quality assurance tool for brachytherapy treatments.

## 1. Introduction

The different solutions for performing in vivo dosimetry have limitations that have slowed down their commercial implementation, and thus, their use in clinical practice [1]. It has been estimated that only around 20–30% of European hospitals using brachytherapy perform in vivo dosimetry [2]. Moreover, afterloader timer errors and patient misidentification can be detected in vaginal-cuff brachytherapy (VCBT) with in vivo dosimetry. In addition, the use of the “read and verify systems” in brachytherapy, a very useful tool for correct patient identification, is very limited, unlike in external radiotherapy [3]. Many of the possible adverse events in brachytherapy can be detected by prospective prevention measures based on some accidents reported in the literature [4]. However, in vivo dosimetry in brachytherapy seems to be the most practical solution because it is able to detect errors not yet reported in the wide literature available [5,6,7].

One of the problems associated with internal in vivo dosimetry in brachytherapy, which is characterised by a high dose gradient, is the size of the detector. It must be: (1) of small size to minimise the effects of dose averaging detection on a volume as much as possible, and (2) compatible with the thin interstitial catheters of the most common brachytherapy applications including prostate [8,9], vaginal-cuff brachytherapy (VCBT), and other gynaecological sites such as cervical cancer [10,11,12]. The use of different in vivo dosimetry systems such as inorganic scintillator detectors, plastic scintillator detectors (PSD), and microMOSFET among others, has been extensively described [9,11,12,13]. PSDs have ideal properties: good spatial resolution, real-time reading, flexibility, linearity with absorbed dose, water equivalence, low temperature, angular and energy dependence, radiation resistance, and repeatability and can be manufactured with sizes smaller than 1 mm [14,15,16,17].

The purpose of this work was to test the ability of a PSD system to measure treatment source dwell times in VCBT of postoperative endometrial carcinoma to increase the quality of brachytherapy treatments in terms of dwell time verification. Great effort has been carried out in this field with plastic and inorganic scintillators inside needles, but to our knowledge. no study on PSD inside bladder catheters or vaginal cylinders has been published [18]. To verify the hypothesis of the present work, five factors that condition dwell time measurement were characterised: (a) repeatability, (b) long-term stability, (c) signal-to-noise ratio, (d) linearity with absorbed dose, and (e) linearity with absorbed dose rate. The results obtained in dwell measurements in treated patients were analysed to verify the ability of the PSD detector in a clinical setting.

## 2. Materials and Methods

The equipment used in this work was the PRO-DOSE (NU-RISE, Ilhavo, Portugal) in vivo dosimetry system that incorporates an organic scintillator BCF-12 (Saint Gobain Crystals, France) with a length of 2 mm attached to a 1.5 m long polymethyl methacrylate (PMMA) optical fibre (TC-500, Asahi Kasei, Tokyo, Japan) with a core diameter of 0.5 mm [19]. The detection technology of the PRO-DOSE system consists of a silicon photomultiplier (SiPM) matrix (Hamamatsu, Japan). The equipment works with an automatic temperature correction and an active cooling system based on a Peltier cell and fans.

The phantom measurements were carried out in a 10 cm × 10 cm cylindric PMMA phantom. The phantom has 4 holes located in the phantom periphery for Ir-192 source insertion through needles of 1.22 mm internal diameter at 4 cm from the phantom axis and one central accessory to insert the PSD probe.

The patient measurements were carried out in 20 sessions of 17 patients undergoing postoperative endometrial carcinoma brachytherapy with an afterloader microSelectron v3 Digital (Elekta, Amsterdam, The Netherlands). The treatment plans were calculated using Oncentra Brachy TPS v.4.5.3 (Elekta, Amsterdam, The Netherlands) after importing the 0.6 mm spaced axial images from a Siemens Somatom go.Open Pro CT (Siemens Healthineers, Erlangen, Germany). Two different types of measurements were carried out with the PSD inserted in: (a) a bladder catheter and (b) at the vaginal cylinder applicator surface.

In the bladder measurements, the PSD probe was modified with the insertion of a gold marker in the tip to visualise the position of the detector in the CT images. In this case, the PSD probe was inserted in a three-way bladder Foley catheter especially designed for dosimetry (Rüsch-Teleflex, Athlone, Ireland) before applicator insertion in the operating room.

For the measurements in VCBT, a cavity was drilled in the vaginal cylinder for the insertion of the PSD probe. Measuring inside an applicator instead of in organs at risk (OAR) offers the main and important advantage of greatly reducing the uncertainty of the PSD probe position because the movement of OAR does not affect the PSD signal measurement. The cylinder was inserted in the vagina in the operating room and secured with elastic bands as reported elsewhere by our group [20,21]. The CT images provide confirmation of good contact of the applicator with the vaginal-cuff and are also the image set for 3D treatment planning with the criteria of vaginal toxicity [21,22,23,24]. All the patients were treated with one of the following two schemes: (a) one session of 7 Gy after external brachytherapy or (b) two fractions of 7.5 Gy in exclusive brachytherapy. The prescription and optimisation points were located at 5 mm of the vaginal mucosa. After evaluation of the EQD2 (α/β = 3) to the most exposed 2 cm^3^ of vagina, a constraint of 68 Gy was established. The position of the PSD was identified in the CT images to obtain the contribution of each dwell position of the source in the detector. In the cases in which the PSD was inserted in the drilled cavity near the surface of the vaginal cylinder, its position was known to be geometrically relative to the applicator. When the special dosimetry bladder catheters were selected for detector probe insertion, the PSD probes used included a gold marker at a known distance from the detector for visual reconstruction in the CT images. After treatment administration, the dwell times for each active source position obtained by the TPS were compared with those measured by the PSD.

### 2.1. Data Analysis

Although the PSD signal can be visualised in real-time, the raw measurements were processed with a MATLAB R2018b (The MathWorks, Inc., Natick, MA, USA) script to remove the SiPM dark noise and to obtain the duration in each dwell position for comparison with the planned values.

In the SiPM operation, the pulses are produced not only by photon-generated carriers but also by thermally generated carriers. The number of dark pulses observed is referred to as dark or background noise. As the noise is produced by thermally-generated carriers, the dark count rate varies with the room temperature. For each measurement, the dark count rate was identified, and a Matlab curve fitting function was applied to interpolate a model to fit the data. The returned model was applied in the raw data to remove the dark noise.

When the afterloader moves the source from one dwell position to another, the measured count rate changes, and it is constant when the source is stopped. The analysis procedure consists of finding abrupt changes through the Matlab function findchangepts (dataset, N) in the signal and identifying the set of data where the source is stopped through the consecutive changes in the absorbed dose rate (transit of the source). The number N of abrupt changes found are predefined and consist of identifying the N points at which the root-mean-square level of the signal changes most significantly.

The integration of each dwell position was calculated using the Matlab function trapz (Y), which calculates the area under the set of discrete data (Y) by breaking the region into trapezoids. The function then adds the area of each trapezoid to compute the total area. The mean of the count rate of the dosimeter response was calculated using the function mean (Y), and each dwell time was calculated by subtracting the two consecutive times in which changes in the absorbed dose rate occur, previously identified by the function findchangepts (dataset, N).

### 2.2. Repeatability

The PRO-DOSE repeatability was tested using an acquisition frequency of 10 Hz with 10 consecutive irradiations of the PSD with the same afterloader channel, different dwell times, and at 4 cm source-to-detector radial distance. The Ir-192 source reference air kerma rate (RAKR) was 23.45 mGy·m^2^·h^−1^ (corresponding to an apparent activity of 215.0 GBq). The air kerma rate constant used for the calculation of apparent activity was 0.1091 mGym^2^h^−1^GBq^−1^ = 4.0367 mGym^2^h^−1^Ci^−1^. The percent absolute deviation from the mean, standard and average deviation, and coefficient of variation were used for the three variables measured: PSD count rate, PSD total counts, and dwell time measurements. The recommendations for a repeatability test, if the measuring device behaviour is approximately linear, is to choose a low and a high value within the range of the measured parameter. In our case, the range of dwell times in postoperative endometrial cancer treatments was around 100 s considering the different Ir-192 source activities, cylinder diameters and prescribed doses. Rarely are source dwell times smaller than 20 s or bigger than 200 s. Therefore, we decided to measure repeatability with 10 s, 30 s, and 300 s dwell times to include all the possible clinical dwell times. Finally, we also tested the system repeatability with a 20 Hz acquisition frequency to verify the capability of the PRO-DOSE with short dwell times up to 1 s for future use of the detector in interstitial cervix brachytherapy, in which low dwell times in the active positions of the needles are usual.

### 2.3. Long-Term Stability

The long-term stability was tested in a range of Ir-192 source reference air kerma rate (RAKR) between 41.19 mGy·m^2^·h^−1^ and 23.65 mGy·m^2^·h^−1^ over a period of two months from 1 November to 31 December 2021 with three consecutive irradiations of the PSD using the same afterloader channels (#1–#4), dwell time (20 s), and source position (at 4 cm from the PSD). The measurements were not corrected by source decay because it takes less than 20 min to complete the test, which represents less than a 0.01% decrease in RAKR.

### 2.4. Signal-to-Noise Ratio

When analysing the data obtained in the long-term stability measurements, the signal-to-noise ratio was calculated as the source activity decreased. The signal amplitude is the signal magnitude (mean count rate) of the high-state level under a fixed irradiation time. The noise is defined as the standard deviation of the data fluctuations. The signal-to-noise ratio (SNR) was analysed considering: (a) the noise measured in the low state level, without radiation, and (b) the noise measured in the high state level, with radiation.

### 2.5. Linearity of PSD Total Counts with Dwell Time (Absorbed Dose)

The PRO-DOSE linearity was tested with a series of increasing dwell times in consecutive irradiations of the PSD ranging from 0.1 s to 204.8 s, which completely covers the clinical dwell times of brachytherapy treatments. The same afterloader channel was used in each case, with a PRO-DOSE acquisition frequency of 10 Hz, and at 4 cm source-to-detector radial distance. The Ir-192 source reference air kerma rate (RAKR) was 27.72 mGy·m^2^·h^−1^.

### 2.6. Linearity of PSD Count Rate with Absorbed Dose Rate

When analysing the data obtained in the long-term stability measurements, the PSD count rate linearity with absorbed dose was tested. The setup and source activities were the same as indicated in the long-term stability section.

### 2.7. Precision of Dwell Time Measurements in Patient Treatments

In the 20 measurements corresponding to 17 women, treated from February 2020 to September 2021, 13 corresponded to bladder Foley catheter measurements, and 4 were measurements in the drilled cylinder. In these 20 fractions, a total of 121 dwell time positions were planned. All the treatment plans had 6 active dwell positions, but one had 7. Two PSD probes were used for these measurements.

### 2.8. Statistical Analysis

For the statistical analysis, the continuous variables were described by their mean, standard deviation, relative deviation, relative standard deviation, and interquartile ranges. The linearity was studied using a linear model. The Lin’s concordance coefficient was used to study the concordance between planned and measured dwell times; the type I error was set to 5%. Analysis was performed using R software for Windows version 4.0.3 (R project for statistical computing; Vienna, Austria).

## 3. Results

In all the measurements described in the following subsections, a short waiting time of 3 min was used to achieve temperature equilibrium of the electronics by the stabilisation Peltier system.

### 3.1. Repeatability

Table 1 summarises the results of the different combinations. See Appendix A. For the 10 Hz acquisition frequency, the maximum time deviation from the nominal dwell time was −0.2 s for the different dwell times measured (10 s, 30 s, and 300 s). The coefficient of variation of the 10 signals is less than 0.5% for the average count rate, 1.4% for the total counts, and 0.8% for the time measured, demonstrating the high repeatability of the PRO-DOSE equipment.

### 3.2. Long-Term Stability

The relative standard deviation of the mean PSD sensitivity with count rate varied less than 1%, demonstrating the high stability of the PRO-DOSE with cumulative absorbed dose and its low absorbed dose rate dependence. Figure 1 shows the sensitivity percent deviation as a function of the absorbed dose rate. The PSD long-term stability as a function of the absorbed dose rate shows mean deviations below 1.5%. A possible explanation for the large standard deviation in the sensitivity measurement corresponding to 0.54 cG/s was that the transfer tube was too close to the PSD probe. This produced a spike in the last dwell position that is typical of these situations. The relative standard deviation of the PSD sensitivity with total counts during the period of long-term stability measurements was lower than 1.2%. See Appendix A.

### 3.3. Signal-to-Noise Ratio (SNR)

The noise was measured in the background (without radiation) and in the signal (with radiation) (Figure 2). As the results in both cases were very similar regardless of the absorbed dose rate, only the noise in the background was used in the analysis.

In Figure 3, the SNR shows an excellent linear relation with the absorbed dose rate.

See Appendix A which summarises the signal-to-noise ratio results obtained during the period of long-term stability analysis.

### 3.4. Linearity of PSD Total Counts with Absorbed Dose

The linear regression analysis showed an excellent correlation of PSD total counts with the absorbed dose rate. See Appendix A. The regression analysis predicted number of counts showed that for nominal absorbed doses above 3 cGy (~6.5 s dwell time with the RAKR used in this test) the predicted total number of counts deviated less than 2% from the measured counts, demonstrating the good total count linearity of the PRO-DOSE equipment with respect to the absorbed dose. See Appendix A. At an absorbed dose rate of 0.47 cGy/s, the slope is 1.05 × 10^7^ photons/Gy.

### 3.5. Linearity of PSD Count Rate with Absorbed Dose Rate

Figure 4 shows the graphical data of the results obtained. The linear regression analysis showed an excellent correlation between the PSD count rate and the absorbed dose rate. The predicted count rate deviated less than 2% from the measured count rate in the whole absorbed dose rate interval, demonstrating the excellent count rate linearity of the PRO-DOSE equipment with respect to the absorbed dose rate. See Appendix A.

### 3.6. Precision of Dwell Time Measurements in Patient Treatments

The absolute values of the deviations between planned and measured dwell times (Δt) are shown in the histogram in Figure 5. The mean and standard deviation of these differences were 0.0 ± 0.2 s.

From 121 dwell positions, 108 were identified and 13 undetectable by the PRO-DOSE system. The detection efficiency was higher than 89%.

The unidentified dwell positions corresponded to two adjacent dwell positions in which the signal difference was below the detection limit and could not be reliably differentiated by the Matlab script (Figure 6).

These cases were also included in a comparison between planned and measured total treatment times. The mean and standard deviation of these differences were 0.2 ± 0.4 s (k = 1). The measured total treatment time deviation in the 20 treatments was always below 0.3%.

The behaviour of the PRO-DOSE system in the whole range of treatment dwell times showed good linear agreement between planned and measured dwell times, see Appendix A. To better visualise the small discrepancies between planned and measured dwell times, their quotient was calculated, showing deviations below 1%, except in one case. Nevertheless, in this case the deviation was below 1.5%, see Appendix A.

Most of the VCBT treatments in our institution use an active length of 2.5 cm with a step size of 0.25 cm. The active loading pattern is one active followed by one inactive dwell position. The active length of 2.5 cm is composed of six active dwell positions. The reason behind this loading pattern is to better quantify the number of dwell positions in the autoradiography performed before each treatment. In Figure 7, the absolute dwell time difference between the planned and the measured value is represented in a box plot for each of these six dwell positions. It can be clearly seen that the median of measured dwell times in the more distal position (#1) is around 0.1 s higher than the planned values.

## 4. Discussion

### 4.1. Repeatability

Precision is a term that describes the repeatability of measurements. Johansen et al. consider a precision of 0.2 s sufficient for detecting dwell time deviations with a clinical impact [25]. The transit doses are not taken into account in the treatment planning system (TPS). During treatment, as the source moves with finite velocity, the transit dose has to be compensated by the afterloader reducing the dwell position doses (by decreasing their corresponding dwell times) to achieve the planned dwell position doses. A high dose rate (HDR) VCBT in endometrial postoperative treatments uses relatively high dwell times, and the effect of time measurement precision is not so critical because transit dose between dwell positions is negligible compared to the dwell position contribution. The excellent repeatability behaviour of the PRO-DOSE will be an advantage in future in vivo measurements in intracavitary (IC)–interstitial (IS) cervical cancer treatments because considerably smaller dwell times are used in the active positions of needles, and the transit and interdwell contributions are considerable [26]. Debnath et al. reported results for the repeatability test using their inorganic scintillator detector with a variation lower than 0.35% in all measurements, with a maximum deviation of 0.54% [27]. The detector response reported by Jorgensen et al. across 11 irradiations conducted for the short-term stability test was 0.6% [28]. The values obtained in their work are very similar to the present results regardless of whether we consider the PSD average count rate, the PSD total counts, or the relative standard deviation of measured dwell time. On the other hand, with an acquisition frequency of 10 Hz, the time average deviation between the nominal and measured PSD dwell time is negative in the whole dwell time range (10 s–300 s). The amount of this mean deviation is between 0.02 s and 0.04 s, coinciding with the reported values [29,30,31,32]. In the irradiation conditions of this work, the present study with 20 Hz acquisition frequency indicates that the PRO-DOSE equipment can reliably measure dwell times as low as 1 s. For lower dwell times that can be generated in IC–IS treatment planning of cervix brachytherapy, the transit time of the source between adjacent dwell positions represents a significant part of the total dwell time and would increase the uncertainty in the absorbed dose distribution; therefore, it would be convenient to avoid such low dwell times [25]. For such short dwell times (1 s), the PRO-DOSE relative standard deviation of the measured dwell times is 4%, which suggests that the best alternative is to set an acquisition frequency above 20 Hz.

### 4.2. Long-Term Stability

A significant decrease in sensitivity with accumulated absorbed dose was observed in a study by Jørgensen et al. with inorganic scintillation detectors (ISD) for pulsed dose rate (PDR) but not for HDR brachytherapy [28]. They reported a standard deviation of the residuals from the linear fit of 1.9% for the probe used in HDR treatments. In our study, the relative standard uncertainty of the residuals was 0.8% for the PSD count rate and 1.1% for the PSD total count stability with accumulated absorbed dose. It is not feasible to separate the sensitivity stability from the linearity with the absorbed dose and the absorbed dose rate of the detector unless a set of sources with different activities is available. Otherwise, measurements should be performed at shorter distances (to maintain the same dose rate) with source decay, introducing a higher positioning uncertainty in the test. From the good stability results obtained, we can infer that the linearity with the absorbed dose and with the absorbed dose rate of the PRO-DOSE system is also around 1%.

### 4.3. Signal-to-Noise Ratio (SNR)

The SNR is an important aspect in signal detection because it compromises the differentiation between adjacent dwell positions. As source activity decays, the signal-to-noise ratio is reduced, as is the capability to differentiate consecutive dwell positions with similar absorbed dose rates. An option in these cases would be to use two detectors as proposed by Guiral et al. [33]. However, the sensitivity of the PRO-DOSE system in terms of the SNR is excellent at the measurement distance (4 cm) from the source for all the range of RAKR. This calibration distance is larger than the separation of the detector from the source in the patient measurements, ensuring that the SNR in these cases is much higher than the values reported in the SNR section. Although Debnath et al. reported a higher SNR for their inorganic scintillator detector, the energy dependence of this type of detector is an important disadvantage in comparison with the PSD in the present study [27].

While the noise in the signal is an important parameter to differentiate consecutive dwell positions, noise in the background would be also relevant in needles with only one active position far away from the PSD. As the signal fluctuations of the PSD measured at the high-state level (signal) and at the low-state level (noise) were constant for all ranges of RAKR (variation of 1.8%), the signal noise of our detector is in practice independent of the source activity or absorbed dose rate in the geometry presented, and it is not affected by the irradiation. This result suggests that the SNR of the PRO-DOSE can be applied to distinguish either consecutive dwell positions or only one active position.

### 4.4. Linearity of PSD Total Counts with Absorbed Dose

Although the linearity of total counts with the absorbed dose does not directly affect dwell time measurements, it is important to keep in mind that in some applications low dwell times can be used. For example, in interstitial cervix brachytherapy applications (in which dwell times well below 10 s are common in the active needle positions) these measurements are conditioned by the acquisition frequency used. For dwell times lower than 6.4 s, to maintain a total count deviation below 2%, a higher acquisition frequency would be necessary to avoid the limitation for the sampling frequency established by the Nyquist theorem. However, as endometrial postoperative HDR treatments with vaginal cylinders use relatively high dwell times (>>6.4 s) compared to the active needle position dwell times in IC-IS cervix applications, the limitation in the linearity of PSD total counts with absorbed dose is not relevant.

### 4.5. Linearity of PSD Count Rate with Absorbed Dose Rate

Jorgensen et al. obtained a percent deviation of around 3% in absorbed dose linearity for their high-Z inorganic scintillator used in HDR brachytherapy [28]. The percent deviation of count rate from the predicted of our PSD detector was below 2% for all the range of source activities reported (similar to the source activity range reported by Jorgensen et al. [28]).

### 4.6. Precision of Dwell Time Measurements in Patient Treatments

The mean of the absolute differences between planned and measured dwell times reported by Johansen et al. was −0.02 s, while a mean of −0.003 s was obtained in the present study. According to our measurements, the standard deviation is 0.21 s, while they reported a value of only 0.06 s. Their higher acquisition frequency is probably the reason for this difference in standard deviation [25].

The dwell times measured with bigger differences (0.55 s) with respect to the planned times may be related to adjacent dwell positions with a small change in terms of count rate, which could cause a bad differentiation between dwell times. Despite these differences of the individual dwell positions, it is important to note that the maximum total treatment time difference was below 0.3% in all the patients measured. The average accuracy reported by Debnath et al. was 0.09%, while the percent deviation of all the measured dwell times in this work was below 0.01% [27].

One of the limitations of this study was the detection efficiency. From the 121 dwell positions measured in 13 cases (10%), the Ir-192 source produced an absorbed dose rate in the PSD, so similar to that in an adjacent dwell position that they could not be differentiated. The stem signal conditions the SNR, and the latter affects the differentiation between adjacent dwell positions. However, we are developing a phantom to characterise the stem signal in terms of angular and distance response to improve the results reported. Although the detection efficiency of the PSD system was only 89%, it is higher than that reported in previous studies [25].

The median dwell time differences between measured and planned dwell times in the first and last dwell position were 0.1 s and 0.060 s in our study, which coincide with results of Fonseca et al. in a recent study [34]. They described differences less than 0.05 s for all dwell positions except for the first and the last position. In the first dwell position, they observed a maximum difference of 0.150 s. This finding suggests that the remote-afterloading brachytherapy system is not correctly considering the compensation of transit time in the more distal dwell position.

The excellent linearity in count rate and total counts with dwell time of the PRO-DOSE system was replicated in the patient measurements supporting the reliability of the use of PSD equipment in clinical settings. The PRO-DOSE system could be useful as a quality assurance tool in terms of dwell time verification for brachytherapy treatments.

## Figures and Tables

**Figure 1 jpm-12-00911-f001:**
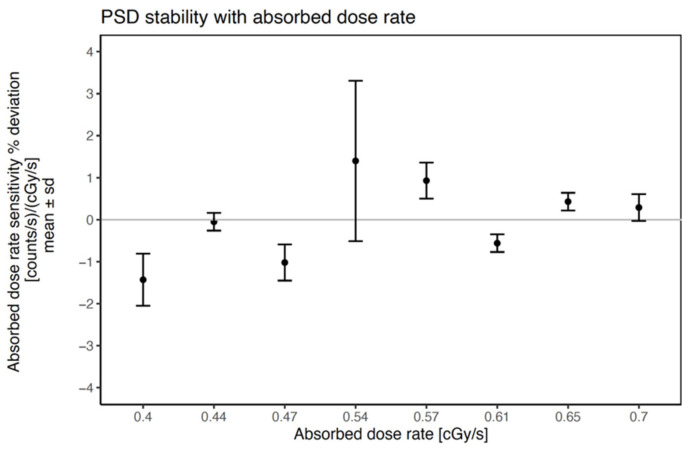
Absorbed dose rate sensitivity percent deviation as a function of absorbed dose rate. Sensitivity is the quotient between PSD count rate and the absorbed dose rate. The black dots indicate the mean value of three consecutive measurements. The error bars are based on standard deviation.

**Figure 2 jpm-12-00911-f002:**
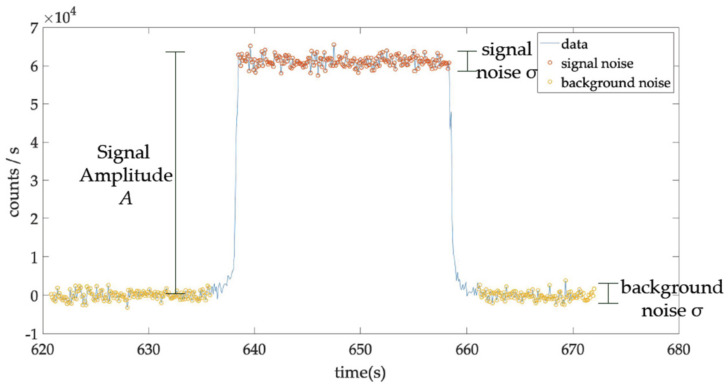
Illustration of the noise in the background and noise in the signal.

**Figure 3 jpm-12-00911-f003:**
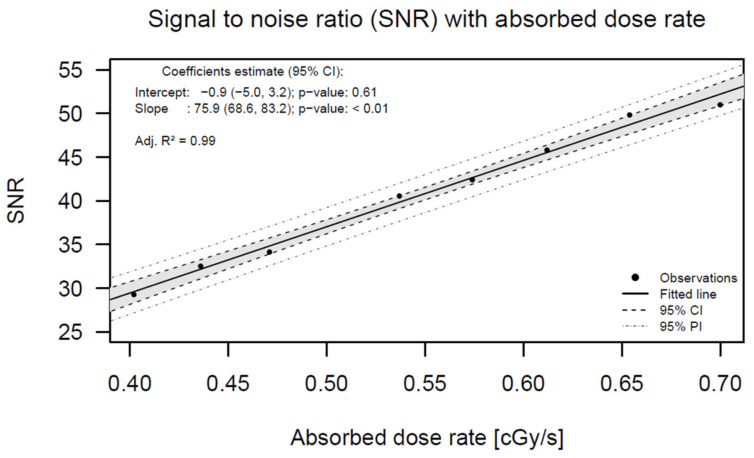
SNR (signal-to-noise ratio); SNR has no units because signal and noise units are the same.

**Figure 4 jpm-12-00911-f004:**
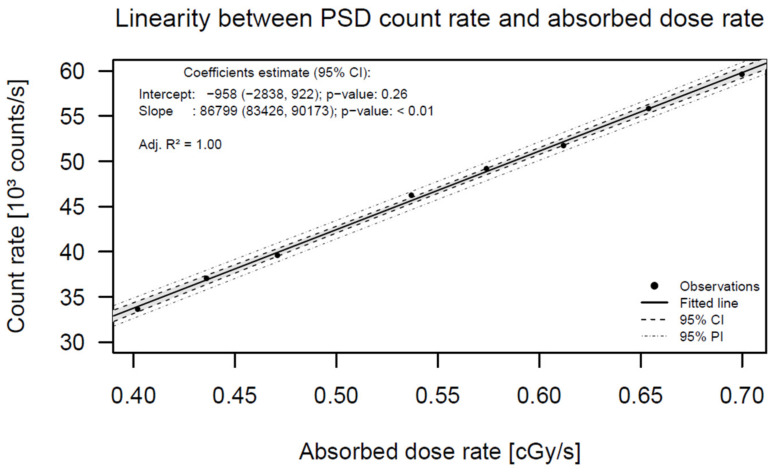
Linear regression of the PSD count rate with the absorbed dose rate.

**Figure 5 jpm-12-00911-f005:**
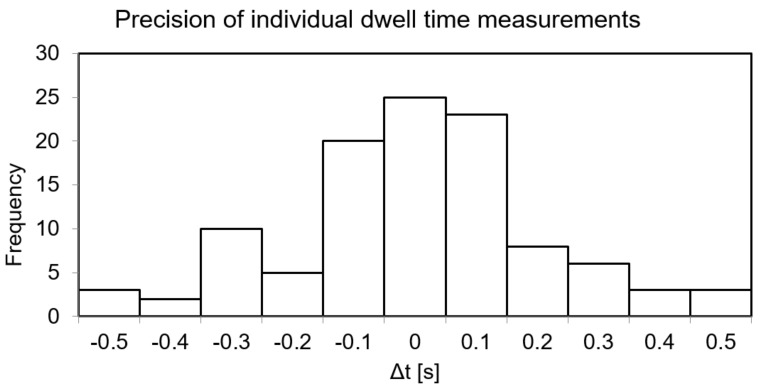
Histogram of the absolute deviations between planned and measured dwell times. The binning is 0.1 s based on the PRO-DOSE acquisition frequency used.

**Figure 6 jpm-12-00911-f006:**
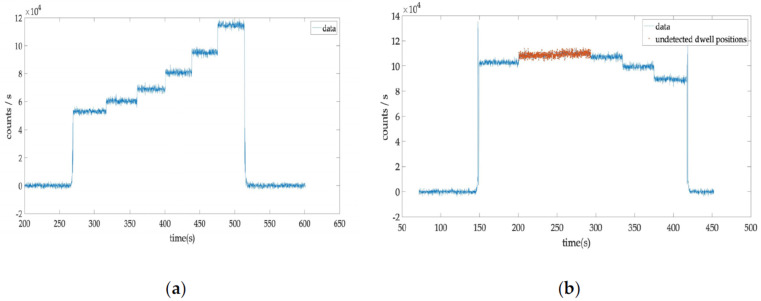
(**a**) An example of six dwell position signals clearly differentiated; (**b**) an example with two dwell positions signals undistinguished due to their similar count rate.

**Figure 7 jpm-12-00911-f007:**
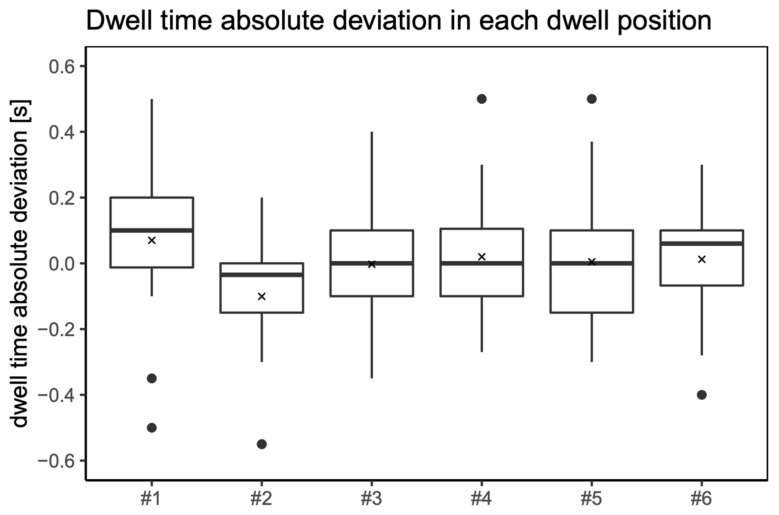
Box plot of the dwell time absolute deviations in each dwell position; #1 is the most distal dwell position, near the tip of the applicator. The step size used in all patients was 2.5 mm, with an inactive source position between active dwell positions: 1-3-5-7-9-11. As only 1 treatment plan used 7 dwell positions (1-3-5-7-9-11-13), the corresponding position number 13 was not included in this box plot. The 6 black dots are outliers: values outside 1.5 times the interquartile range above the upper quartile and below the lower quartile. The average values in each of the 6 dwell positions are represented by the crosses and are from left-to-right: 0.070 s, −0.100 s, −0.003 s, 0.020 s, 0.005 s, and 0.012 s, and the corresponding medians are: 0.100 s, −0.035 s, 0.000 s, 0.000 s, 0.000 s, and 0.060 s.

**Table 1 jpm-12-00911-t001:** Summary of the results of the repeatability test.

Detector Acquisition Frequency [Hz]	Nominal Dwell Time [s]	Time Average Deviation [s]	Maximum Time Deviation [s]	RSD ^1^ Count Rate [%]	RSD ^1^ Total Counts [%]	RSD ^1^ Measured Dwell Time [%]
10	300	−0.04	−0.10	0.27	1.17	0.02
	30	−0.02	−0.10	0.22	0.29	0.14
	10	−0.02	−0.20	0.50	1.38	0.79
20	10	0.00	−0.10	0.32	5.59	0.58
	3	0.06	0.15	0.43	2.45	1.89
	1	0.04	0.10	1.03	9.22	4.19

^1^ RSD: relative standard deviation.

## Data Availability

The data presented in this study are available upon request from the corresponding author.

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
