# Peer review of "In Vivo Verification of Treatment Source Dwell Times in Brachytherapy of Postoperative Endometrial Carcinoma: A Feasibility Study"

_jpm, 2022, doi:10.3390/jpm12060911_

Round 1
Reviewer 1 Report
I would like to express my pleasure to evaluate your article. I think in vivo dosimetric studies are very important. It will contribute to our routine practice.
This study is aiming to evaluate the performance of a PSD in postoperative endometrial brachytherapy in terms of source dwell time accuracy.
The authors reported that 1% standard deviation as a result of the test. Also they pointed out excellent PSD sensitivity and stability. The most important result of this study is the mean (standard deviation) of the absolute differences between planned and measured dwell times in patient treatments is 0.0 (0.2) seconds.
I think that the authors addressed all issues about in vivo plastic scintillator dosimetry.
Author Response
Response to Reviewer 1 Comments
Point 1: I would like to express my pleasure to evaluate your article. I think in vivo dosimetric studies are very important. It will contribute to our routine practice.
This study is aiming to evaluate the performance of a PSD in postoperative endometrial brachytherapy in terms of source dwell time accuracy.
The authors reported that 1% standard deviation as a result of the test. Also they pointed out excellent PSD sensitivity and stability. The most important result of this study is the mean (standard deviation) of the absolute differences between planned and measured dwell times in patient treatments is 0.0 (0.2) seconds.
I think that the authors addressed all issues about in vivo plastic scintillator dosimetry.
Response 1: Thank you very much for your review and for recognizing our efforts, we are also convinced that routine in vivo dosimetry will be a standard and a quality indicator in brachytherapy.
Reviewer 2 Report
I faced an article, which is very interesting about checking the possible uncertainty in Brachytherapy treatments. Also it seems has own uncertainty.
I have no idea about the journal's policy about publishing some images in the articles, but I think your manuscript needs many images for better understandings.
Reviewer 3 Report
Title:
In vivo verification of treatment source dwell times in brachytherapy of postoperative endometrial carcinoma: a feasibility study
Review
The article presents an in vivo procedure for the verification of dwell times in a clinical brachytherapy application using a commercially available plastic scintillator based dosimetric system. The applied simple clinical application consisting of one catheter and the fact that only the dwell times were verified, reduces the scientific importance of the current work. However, the relatively large number of experiments and the corresponding given results renders the work potentially publishable, provided that the following comments have appropriately addressed:
1. Line 30 (“…maintained an excellent long-term stability...”). Please, be quantitative rather than qualitative in your statement.
2. Line 42. Please correct the miss type “slowed” should be “showed”
3. Lines 45 – 50: The authors give in this section a general description of the procedures that should be checked in an in vivo brachytherapy procedure. Most of the stated procedures, however, do not apply for the chosen brachytherapy application. Can the authors clarify which errors can be checked using in vivo measurement of source dwell times? Specifically, in the VCBT treatments used for clinical measurements in this work, there is zero probability of erroneous connection since a single channel is used. This severely limits the scope of this work since source dwell time verification can be performed using routine QA methods that are less demanding (in all aspects of time, effort and cost) relative to in vivo measurement. Hence the main offering of this work is the characterization of the system used by the authors. Are these results new and system/implementation specific?
4. Lines 61-63: This disregards in vivo dosimetry approaches using detectors outside the patient (e.g., https://doi.org/10.1118/1.4823758 or https://doi.org/10.1088/1361-6560/ac612d)
5. Lines 72-73: Please refer to comment on lines 45-50 and be more specific.
6. Line 75 (“…inside bladder catheters or vaginal cylinders…“). How do these applications differ and why is it interesting to report corresponding results?
7. Lines 77-78. Linearity with absorbed dose is an established property of PSDs according to lines 66-67 above. The appropriateness of PSD sensitivity for brachytherapy IVD has also been documented in the literature.
8. Line 122-123 (“The position of the PSD was identified..”). Please elaborate. Was this identified visually or geometrically given the known position of the PSD on the applicator?
9. Line 130 (“… raw measurements …”). The stem effect is known to be an important confounding factor to measurements with PSDs. Was it corrected and if not how is it expected to vary with irradiation conditions?
10. Lines 146-148 (“The number N of ….. significantly”). Please revise for clarity since the number of change points to report, N, is an input. What would happen in case of erroneous transfer tube connection leading to a number of source dwells different than expected (although this does not apply to the clinical scenario studied in this work)?
11. Line 228 (“..type I error was set to 5%.”). Can the authors provide a reason for setting the type I error to 5% ?
12. Table 1 (Relative standard deviation total counts column). While one would intuitively expect that counting statistics improve with dwell time, results are worse for 300 s and 10 s relative to 30 s and 3 s, respectively. Please comment.
13. Figure 1. Please recast the x axis in terms of dose rate.
14. Figure 1 – legend lines 259-262 (“The error bars …. graph.”). This belongs to the discussion of the figure and not the caption. Please explain how this differs from a transit time effect.
15. Lines 301-302 (“The mean and …. 0.2 s”). How does this compare to the 0.1 s accuracy set by the IEC Part 2-17 (Particular requirements for the basic safety and essential performance of automatically-controlled brachytherapy afterloading equipment)?
16. Line 306 (“.correctly identified”). What was the criterion for "correct identification"? Probably "correctly identified" should be replaced by "identified".
17. Line 310 (“..detection limit..”). What is this detection limit or, at least, what does it depend on?
18. Line 349. Is this consistent with results in Table 1? What column of this table does this result correspond to?
19. Lines 350-352. Please substantiate this statement by reference to the way transit dose is handled by the aterloader used.
20. Lines 366 – 367. Surely this statement applies to the irradiation conditions of this work. Please make sure you indicate this.
21. Lines 447-448. What is the acquisition frequency range of your method and why didn't the authors use a higher frequency?
22. Lines 478-479. This is a very general statement. Please refer to comment on lines 45-50.
Author Response
Please see the attachment.
Response to Reviewer Comments:
First of all, thank you very much for your excellent review, it will considerably improve the quality of the manuscript. We have tried to address these issues as thoroughly as we can and hope you will find them satisfying.
Point 1: Line 30 (“…maintained an excellent long-term stability...”). Please, be quantitative rather than qualitative in your statement.
Response 1: Thank you very much for your valuable comment. We have made the following modification to this sentence in the abstract section: “The relative standard deviation of the PSD sensitivity with accumulated absorbed dose was lower than 1.2%.”
Point 2: Line 42. Please correct the miss type “slowed” should be “showed”.
Response 2: Thank you very much for your suggestion, but our intention was to explain that the limitations of the different systems for performing in vivo dosimetry have slowed down their commercialization, and not that these limitations have been shown in the commercial implementations. We have added the preposition “down” to the verb to correct this point. The modified sentence now reads: “The different solutions for performing in vivo dosimetry have limitations that have slowed down their commercial implementation, and thus, their use in clinical practice [1].”
Point 3: Lines 45 – 50: The authors give in this section a general description of the procedures that should be checked in an in vivo brachytherapy procedure. Most of the stated procedures, however, do not apply for the chosen brachytherapy application. Can the authors clarify which errors can be checked using in vivo measurement of source dwell times? Specifically, in the VCBT treatments used for clinical measurements in this work, there is zero probability of erroneous connection since a single channel is used. This severely limits the scope of this work since source dwell time verification can be performed using routine QA methods that are less demanding (in all aspects of time, effort and cost) relative to in vivo measurement. Hence the main offering of this work is the characterization of the system used by the authors. Are these results new and system/implementation specific?”.
Response 3: Thank you very much for your helpful remark. You are right, we did not specifically indicate the VCBT manual procedures susceptible to error (a possible error could be patient misidentification, because we usually treat more than one patient/day). Instead, we give a set of general manual procedures more related to brachytherapy treatments with non-indexed catheters. The reason was that we wanted to highlight the main arguments why in vivo dosimetry is necessary in brachytherapy. Our objective is to implement in vivo dosimetry in IC/IS cervix brachytherapy and multichannel cylinder treatments in the future, so we planned a first approach in a simpler scenario (single channel VCBT) limiting our study to source dwell time verification. As you rightly comment, we have centered our efforts in this work in the characterization of the system in terms of source dwell time verification. We have made the following modification to this sentence: “Moreover, afterloader timer errors and patient misidentification could be detected in vaginal-cuff brachytherapy (VCBT) with in vivo dosimetry.”
Point 4: Lines 61-63: This disregards in vivo dosimetry approaches using detectors outside the patient (e.g., https://doi.org/10.1118/1.4823758 or https://doi.org/10.1088/1361-6560/ac612d)
Response 4: Thank you very much for your corrections and suggestions. We have made the following modification to the sentence: “One of the problems associated with internal in vivo dosimetry in brachytherapy, which is characterised by a high dose gradient, is the size of the detector.”
Point 5: Lines 72-73: Please refer to comment on lines 45-50 and be more specific.
Response 5: Thank you very much for your valuable comment. We have made the following modification to the sentence: “The purpose of this work was to test the ability of a PSD system to measure treatment source dwell times in VCBT of postoperative endometrial carcinoma to increase the quality of brachytherapy treatments in terms of dwell times verification”.
Point 6: Line 75 (“…inside bladder catheters or vaginal cylinders…“). How do these applications differ and why is it interesting to report corresponding results?
Response 6: Thank you very much for your questions, because we did not clarify these two approaches of PSD insertion. Using the bladder catheters was the first attempt, but then the bladder catheter for radiation measurement (Rüsch/Teleflex, Wayne, PA) was commercially discontinued. In addition, we thought that we could have more robust results in terms of dose rate measurement if we used a hole in the vaginal cylinders instead of the bladder catheter to insert the PSD, because the relative position source-PSD is less stable due to the internal motion of the OAR and its filling. In terms of source dwell time measurement, there were no significant differences between the two approaches, so we did not add any additional explanations.
Point 7: Lines 77-78. Linearity with absorbed dose is an established property of PSDs according to lines 66-67 above. The appropriateness of PSD sensitivity for brachytherapy IVD has also been documented in the literature.
Response 7: Thank you very much for your comments. Again you are right, “linearity with absorbed dose is an established property of PSDs”, but we verified our own system linearity in terms of (a) total counts (absorbed dose) and (b) count rate (absorbed dose rate). The same approach was used for the sensitivity stability, testing it again in terms of both magnitudes. In our opinion this is a novel presentation not explicitly reported in the literature.
Point 8: Line 122-123 (“The position of the PSD was identified..”). Please elaborate. Was this identified visually or geometrically given the known position of the PSD on the applicator?
Response 8: Thank you very much for your helpful remarks. We have added two sentences with an explanation of PSD position reconstruction: “In the cases in which the PSD was inserted in the drilled cavity near the surface of the vaginal cylinder its position was known to be geometrically relative to the applicator. When the special dosimetry bladder catheters are selected for detector probe insertion, the PSD probes used included a gold marker at a known distance from the detector for visual reconstruction in the CT images.”
Point 9: Line 130 (“… raw measurements …”). The stem effect is known to be an important confounding factor to measurements with PSDs. Was it corrected and if not how is it expected to vary with irradiation conditions?
Response 9: Thank you very much for your valuable comment. The actual version of the PRO-DOSE software allows PSD signal visualisation in real-time, but the raw signal is not corrected for stem effect. In our next study about PSD calibration (lines 462-464), we are planning to make two sets of measurements with two probes (with and without scintillator) in a water phantom to have a complete along-and-away table, in different angular positions and characterize the amount of stem effect contribution.
Point 10: . Lines 146-148 (“The number N of ….. significantly”). Please revise for clarity since the number of change points to report, N, is an input. What would happen in case of erroneous transfer tube connection leading to a number of source dwells different than expected (although this does not apply to the clinical scenario studied in this work)?
Response 10: Thank you very much for your comment. Although the PSD signal can be visualised in real-time, the data analysis was done after treatment delivery and not in real-time. The ability of the system and data signal processing to identify different catheters, dwell positions and duration of dwell times is shown, but we confirm that it requires the N input (the planned number of dwell positions inside of the catheter). This N value is obtained from the DICOM file. To apply this method in real-time and in a clinical scenario, the treatment plan (DICOM file) would be used to know in advance the following inputs for the real-time data processing: number of catheters, number of dwell positions and corresponding dwell times, and planned doses at each instant of the treatment. The real-time data of the PRO-DOSE would be compared against the treatment plan info in real-time. It can detect drastic errors, such as deviations in treatment dose delivered and erroneous transfer tube connections. If the PSD is positioned in organs at risk, it can also verify if the dose delivered is within acceptable clinical tolerance.
Point 11: Line 228 (“..type I error was set to 5%.”). Can the authors provide a reason for setting the type I error to 5% ?
Response 11: The reason for setting the type I error to 5% is because we are willing to accept a 5% probability that we are wrong when the null hypothesis is rejected. We don’t reduce our risk of committing a type I error (for example: 1%) because it will be less likely to detect a true difference from one that really exists (thus risking a type II error).
Point 12: Table 1 (Relative standard deviation total counts column). While one would intuitively expect that counting statistics improve with dwell time, results are worse for 300 s and 10 s relative to 30 s and 3 s, respectively. Please comment.
Response 12: Thank you very much for your helpful remarks. We checked the data and results and can find no explanation for the difference. The measurements were all done on the same day.
Point 13: Figure 1. Please recast the x axis in terms of dose rate.
Response 13: We have recast the abscissa in terms of absorbed dose rate following your recommendation. We also modified references to activity in the manuscript accordingly.
Point 14: Figure 1 – legend lines 259-262 (“The error bars …. graph.”). This belongs to the discussion of the figure and not the caption. Please explain how this differs from a transit time effect.
Response 14: Thank you very much for your helpful remark. We moved this comment from the legend lines to the main text following your indications. In the graph below (corresponding to a source activity of 7.8 Ci in Figure 1 of the manuscript shows three consecutive measurements in the following conditions: one active dwell position in each of the four needles), we see spikes when the source enters and leaves the 3rd and 4th needle. This only happened in one measurement. We suspect that these transfer tubes were very close to the PSD probe which caused these spikes when the source moved at the transfer tube to the dwell position and when it returned to the afterloader. While the duration of these spikes is about 100 ms (with a frequency of 10 Hz it represents one single acquisition), the transit time duration when the source enters the catheter and leaves it is considerably higher (800 ms as shown in the figure below ) and accordingly to the nominal speed of the source provided by Nucletron: 500 mm/s.
Additional figures for the reviewer’s Point 14: raw data vs. measurement without the spikes (ignored) and below transit time figure.
+
Point 15: Lines 301-302 (“The mean and …. 0.2 s”). How does this compare to the 0.1 s accuracy set by the IEC Part 2-17 (Particular requirements for the basic safety and essential performance of automatically-controlled brachytherapy afterloading equipment)?
Response 15: Thank you very much for your instructive comment. The accuracy of our system when working with an acquisition frequency of 10 Hz is probably not as good as the IEC standard requirements for the accuracy of afterloader timers. However, the objective of dwell time verification is to identify clinically relevant errors (fault in the equipment timer or patient misidentification, for example).
Point 16: Line 306 (“.correctly identified”). What was the criterion for "correct identification"? Probably "correctly identified" should be replaced by "identified".
Response 16: Thank you very much for your helpful remark. The dwell positions were identified or undetected by the algorithm. As you have pointed out, the redundant expression “correctly identified” makes no sense and, therefore, we have eliminated the adverb “correctly” following your comment.
Point 17: Line 310 (“..detection limit..”). What is this detection limit or, at least, what does it depend on?
Response 17: The undetected dwell positions are consecutive dwell positions with a very similar mean signal. This never occurred in the first dwell position (the closest dwell position to the afterloader), because the signal-to-noise ratio of the system was always enough to differentiate between background and a dwell position. The detection limit depends on the findchangepts(dataset, N) and its capacity to identify all the dwell positions. The number N of abrupt changes found is predefined and consists of identifying the N points at which the root-mean-square level of the signal changes most significantly. If the function doesn’t return N abrupt changes, there are undetected dwell positions. This is confirmed by the graph, as shown in Figure 6b of the manuscript.
Point 18: Line 349. Is this consistent with results in Table 1? What column of this table does this result correspond to?
Response 18: Thank you very much for your clever observation. The standard deviation of the measured time results was not reported in the tables of the manuscript. In the supplement material, the first 6 tables correspond to the repeatability test for each dwell time-frequency combination. The standard deviation can be determined from the column “PSD measured dwell time”. We decided to include this value in the discussion section for comparison with the value published by Johansen et al. However, we understand that if the standard deviations are not reported in the tables we have to remove this sentence.
Point 19: Lines 350-352. Please substantiate this statement by reference to the way transit dose is handled by the afterloader used.
Response 19: Thank you very much for your valuable comment. We have added one sentence before this statement: “The transit doses are not taken into account in the treatment planning system (TPS). During treatment, as the source moves with finite velocity, the transit dose has to be compensated by the afterloader reducing the dwell position doses (by decreasing their corresponding dwell times) to achieve the planned dwell position doses.”
Point 20: Lines 366 – 367. Surely this statement applies to the irradiation conditions of this work. Please make sure you indicate this.
Response 20: Thank you very much for your valuable comment. We have completed the sentence following your indications: “In the irradiation conditions of this work, the present study with 20 Hz acquisition frequency indicates that the PRO-DOSE equipment can reliably measure dwell times as low as 1 s.”
Point 21: Lines 447-448. What is the acquisition frequency range of your method and why didn't the authors use a higher frequency?
Response 21: As we didn’t find a significant improvement of the dwell time precision by increasing the frequency during the phantom measurements, we maintained 10 Hz for the patient’s measurements. More studies are necessary.
Point 22: Lines 478-479. This is a very general statement. Please refer to the comment on lines 45-50.
Response 22: Thank you very much for your valuable comment. We agree with you that “quality assurance tool” is a very general statement. We have modified the sentence following your indications: “The PRO-DOSE system could be useful as a quality assurance tool in terms of dwell times verification for brachytherapy treatments.”
Thank you for the instructive comments and your time!

Reviewer 4 Report
This paper describes the characterization of a plastic scintillator-based detector for applications concerning in vivo dosimetry in brachytherapy of postoperative endometrial cancer. Particular attention is dedicated to the measurement the insertion times of the sources in their locations (dwell times), for which the authors have developed a specific script in Matlab environment.
This point is clearly important to implement control systems that guarantee the correct execution of the treatment plan and therefore the effectiveness of the treatment and the safety of the patient. Therefore the study is significant and relevant to the community of medical physicists who develop technological applications of this type in an industrial setting or use them in clinical practice.
Certainly, the control of dwell times is subject to ambiguities of geometric origin (locations with the same dose rate at detector) and is critical in the case of short stationing of the sources, where transit times become important. These issues emerge clearly in the tables and graphs produced by the authors. However, from the study of the material presented and derived from both phantom and in vivo measurements, the detector appears to be well characterized in its properties relevant for the mentioned applications
and the proposed clinical use of the system appears convincing.
Author Response
Response to Reviewer Comments:
Point 1: This paper describes the characterization of a plastic scintillator-based detector for applications concerning in vivo dosimetry in brachytherapy of postoperative endometrial cancer. Particular attention is dedicated to the measurement the insertion times of the sources in their locations (dwell times), for which the authors have developed a specific script in Matlab environment.
This point is clearly important to implement control systems that guarantee the correct execution of the treatment plan and therefore the effectiveness of the treatment and the safety of the patient. Therefore the study is significant and relevant to the community of medical physicists who develop technological applications of this type in an industrial setting or use them in clinical practice.
Certainly, the control of dwell times is subject to ambiguities of geometric origin (locations with the same dose rate at detector) and is critical in the case of short stationing of the sources, where transit times become important. These issues emerge clearly in the tables and graphs produced by the authors. However, from the study of the material presented and derived from both phantom and in vivo measurements, the detector appears to be well characterized in its properties relevant for the mentioned applications and the proposed clinical use of the system appears convincing.
Response 1: Thank you very much for recognizing our efforts and for your review. We are also convinced that in vivo dosimetry is an key element in and the quality of our treatments.